# Aptamers and Antisense Oligonucleotides for Diagnosis and Treatment of Hematological Diseases

**DOI:** 10.3390/ijms21093252

**Published:** 2020-05-04

**Authors:** Valentina Giudice, Francesca Mensitieri, Viviana Izzo, Amelia Filippelli, Carmine Selleri

**Affiliations:** 1Department of Medicine, Surgery and Dentistry “Scuola Medica Salernitana”, University of Salerno, Baronissi, 84081 Salerno, Italy; f.mensi90@gmail.com (F.M.); vizzo@unisa.it (V.I.); afilippelli@unisa.it (A.F.); cselleri@unisa.it (C.S.); 2Unit of Clinical Pharmacology, University Hospital “San Giovanni di Dio e Ruggi D’Aragona”, 84131 Salerno, Italy

**Keywords:** aptamers, antisense oligonucleotides, diagnosis, treatment, hematology

## Abstract

Aptamers or chemical antibodies are single-stranded DNA or RNA oligonucleotides that bind proteins and small molecules with high affinity and specificity by recognizing tertiary or quaternary structures as antibodies. Aptamers can be easily produced in vitro through a process known as systemic evolution of ligands by exponential enrichment (SELEX) or a cell-based SELEX procedure. Aptamers and modified aptamers, such as slow, off-rate, modified aptamers (SOMAmers), can bind to target molecules with less polar and more hydrophobic interactions showing slower dissociation rates, higher stability, and resistance to nuclease degradation. Aptamers and SOMAmers are largely employed for multiplex high-throughput proteomics analysis with high reproducibility and reliability, for tumor cell detection by flow cytometry or microscopy for research and clinical purposes. In addition, aptamers are increasingly used for novel drug delivery systems specifically targeting tumor cells, and as new anticancer molecules. In this review, we summarize current preclinical and clinical applications of aptamers in malignant and non-malignant hematological diseases.

## 1. Introduction

Malignant and non-malignant hematological disorders are a heterogeneous group of clinical conditions including a broad spectrum of malignancies such as leukemias, lymphomas, myeloma, and other myeloproliferative or lymphoproliferative disorders, as well as clonal and nonclonal benign diseases, such as some myelodysplastic syndromes, chronic granular lymphocyte disorders, monoclonal gammopathies, mastocytosis, paroxysmal nocturnal hemoglobinuria, and other bone marrow (BM) failure syndromes [1]. These clinical entities are derived from neoplastic transformation of myeloid or lymphoid cells at various stages of differentiation forming the so-called “clone”, a tumor subpopulation responsible for long-lasting maintenance of the disease, or from somatic mutations, or aberrant immune responses [1,2]. Hematological disorder classification has been recently revised by the World Health Organization in 2016 based on significant progress in our understanding of molecular defects underlying all these diseases and the recognition of new clinical independent entities [2]. Therefore, better diagnostic criteria, differential diagnosis markers, risk stratification, and prognostic definition are required to improve clinical management and quality of life of hematological patients; thus, the discovery of new biomarkers is essential for a better understanding of disease biology, and also for the identification of new possible therapeutic targets. The emerging of highly specific molecular and immunological diagnostic technologies helps hematologists to increase their knowledge of disease pathophysiology and to improve clinical management of patients [1]. Aptamers short three-dimensional RNA or single-stranded DNA oligonucleotides or antisense oligonucleotides (ASOs) can interact to a variety of targets with high affinity and specificity, and thus can be used for analytical or diagnostic applications in the hematologic field [3,4,5,6,7]. In this review, we summarize current applications of aptamers and antisense oligonucleotides in malignant and non-malignant hematological diseases.

## 2. Types of Oligonucleotides

### 2.1. Antisense Oligonucleotides

ASOs are short (12–30 nucleotides in length) chemically synthesized single-stranded oligonucleotides that can modify stability and translation of target complementary mRNAs by specific Watson–Crick base pairing binding [4,8]. ASOs interfere with protein translation through two main processes, i.e., occupancy only or steric blocking and RNA-degradation mechanisms [8]. In the first case, ASOs bind mRNAs preventing interactions with ribosomes and protein translation; however, ASOs are more frequently designed to bind microRNAs (miRNAs), small non-coding RNAs of about 22 nucleotides, and to increase protein expression especially for therapeutic purposes [4,8]. Because miRNAs reduce post-translational protein expression by binding to target mRNAs, miRNA-ASO complexes are not able to interact with mRNAs leading to increase translation of targets. ASOs could also specifically enhance translation of given proteins by directly binding to their 5′-untranslated region (UTR) regulatory open reading frame or stem-loop structure sequences [8,9,10]. Moreover, ASOs can cause RNA splicing alterations such as exon skipping or inclusion or can increase gene expression by blocking gene repression mediated by the interaction between long non-coding RNAs and chromatin [8,11]. The second mechanism of action of ASOs is the mRNA degradation by endogenous nucleases, such as RNase H1 and argonaute 2 (Ago2) [4,8]. RNase H1 is activated by double-stranded RNA complexes or an RNA-DNA heteroduplex causing mRNA cleavage [8]; whereas Ago2 recognizes RNA duplexes in the cytoplasm and degrades the passenger strand at the 5′-end or at the least stable end [8,12].

ASOs are under investigation for treatment of several hematologic disorders (Table 1) [4]. The ASO BP1001 targeting the translation initiation site of the growth factor receptor-bound protein 2 (Grb2) has been tested in acute myeloid leukemia (AML) and high-risk myelodysplastic syndromes (MDS) and showed a good safety profile and evidence of antileukemic activity in a phase I study [13]. Grb2 is downstream of various oncogenic tyrosine kinases and participates in several signaling pathways, such as mitogen-activated protein kinase, breakpoint cluster region protein-Abelson murine leukemia viral oncogene homolog (BCR-ABL), KIT, fms related tyrosine kinase 3 (FLT3), and Janus kinase 2 (JAK2) tyrosine kinases, and is frequently overactivated in cancers [13,14]. ASOs that target Grb2 can exert antileukemic activity by inhibiting Son of Sevenless (SOS) guanine nucleotide exchange mediated extracellular signal-regulated kinases 1/2 (ERK1/2) activation [13]. BP1001 in monotherapy or BP1001 in combination with a low-dose cytarabine have shown efficacy in reducing the frequency of peripheral blood and (BM) blasts [13,15]. Two other phase I and II ongoing clinical trials are evaluating the efficacy of BP1001 in Philadelphia chromosome + AML and chronic myeloid leukemia (CML) in combination with dasatinib, and in AML and high-risk myelodysplastic syndromes (MDS) in combination with decitabine (Clinicaltrials.gov identifiers, NCT02923986 and NCT02781883). Another promising ASO is the anti-BCL2 (B-cell lymphoma 2) oblimersen which has shown synergistic antitumor effects in combination with standard chemotherapeutic protocols for the treatment of various B cell non-Hodgkin lymphomas, such as follicular or diffuse large B cell lymphomas, and multiple myeloma [4,16,17,18,19,20]. BP1002 and SPC2996 also target Bcl-2 in advanced non-Hodgkin lymphoma and chronic lymphocytic leukemia (CLL) (Table 1). Other promising ASOs under clinical investigation are the following: cenersen against TP53 in MDS; the anti-c-myb G4460 for CML treatment in both chronic and accelerated phase; the anti-X-linked inhibitor of apoptosis protein (XIAP) AEG35156 for relapsed or refractory CLL and indolent B cell lymphomas, AML in combination with standard chemotherapy, or refractory/relapsed AML [21,22]; the anti- Hypoxia-inducible factor(HIF)-1α EZN-2968 and the anti- signal transducer and activator of transcription(STAT)3 AZD9150 and IONIS-STAT3Rx for non-Hodgkin lymphoma treatment (Table 1) [4].

### 2.2. Aptamers and Modified Aptamers

Aptamers, or chemical antibodies, are single-stranded oligonucleotides, either DNA or RNA, that bind: proteins by recognizing tertiary or quaternary structures as antibodies, and not primary sequences; or small molecules [4,23,24]. In Table 2, we report aptamers tested in vitro or in vivo for therapeutic and diagnostic purposes but not approved for routine clinical practice, whereas in Table 3, aptamers under clinical investigation for treatment of hematologic diseases are presented (Table 2 and Table 3). Aptamers are produced in vitro through a process known as SELEX, in which short oligonucleotides with fixed end sequences from large random libraries (10^15^ molecules) are amplified by polymerase chain reaction (PCR) and incubated with target molecules (Figure 1A). Subsequently, unbound or weakly bound sequences are removed, aptamers are released from aptamer-target molecule complexes, and sequences are amplified by PCR or reverse transcription. The pool of aptamers is incubated again with target molecules, and unbound or weakly bound sequences removed for a total of eight to twelve rounds of incubation/PCR amplification [23]. After each round, aptamers with increased affinity are selected, amplified, cloned, and sequenced. However, RNA-based aptamers are very sensitive to nuclease- and alkaline condition-mediated degradation because of the nucleophile hydroxyl group at the 2′-position of ribose that increases the hydrolysis rate at the adjacent phosphodiester bound. The use of modified nucleotides can dramatically increase resistance to RNases and is still suitable for T7 RNA polymerase incorporation and PCR amplification [23,24]. Modifications include replacement with 2′-amino or 2′-fluoro nucleotides, and 2′-*O*-methyl group substitution; however, the latter allows selection of high affinity aptamers, but transcriptional yield is very low because of poor PCR amplification [24]. Higher aptamer stability can be achieved using single-stranded DNA (ssDNA) oligonucleotides with modifications at the 5′-position of deoxyuridine providing resistance to DNases. In addition, these modified single-stranded aptamers or SOMAmers bind to target molecules during the SELEX process with less polar and more hydrophobic interactions showing fewer hydrogen bonds and charge–charge interactions as compared with standard aptamers, and slower dissociation rates (> 30 min) while avoiding covalent and permanent bindings associated with high dissociation constant (K_d_) values [23,24]. Indeed, a permanent binding does not allow to remove aptamers from the target for next SELEX rounds, and for further cloning and sequencing. The cell-SELEX process is another way for selection of aptamers using cell lines (Figure 1B). In particular, ssDNA libraries are incubated with cells known to express targeted protein (e.g., CD30 on Hodgkin lymphoma cell lines), and sequences bind with various affinity those targets. Unbound sequences are washed away, while bound sequences are first eluted by heating and subsequently incubated with cells not expressing targeted proteins for counterselection. Afterwards, unbound sequences, which do not recognize non-specific targets on negative cells, are later reverse transcript and amplified by PCR [3]. However, cell-SELEX requires suitable cell lines expressing or not the targeted proteins for positive and negative selection, respectively, and aptamer specificity remains lower than that obtained by SELEX process [4].

Aptamers and SOMAmers are largely employed for proteomics analysis, although mass spectrometry is the standard for in-depth analysis and antibody-based assays are sensitive even for low-abundance proteins, but multiplexing is limited. The SOMAscan assay, a multiplex high-throughput platform, allows the simultaneous measurement of thousands of proteins with high reproducibility showing a median intraplate coefficient of variability (CV) of 3%–4% [25,26]. Moreover, data from different runs can be analyzed together using calibrators and quality control samples for data normalization with a good intra- and inter-subject variability for most analytes [25]. Measurement of protein levels with the SOMAscan assay is also very sensitive with a median lower limit of quantification of 1 pM and for some proteins as low as 100 fM [23]. SOMAmers for proteomics analysis are conjugated to a fluorophore, which is also conjugated to a photocleavable linker and to biotin. SOMAmers are first captured on streptavidin-coated beads (Catch-1), and then incubated with target molecules or biological samples allowing the formation of SOMAmer-target protein complexes. After incubation, complexes are released from beads by disrupting the photocleavable linker with UV light, and unbound or weakly bound SOMAmers are washed away using a polyanionic competitor-containing buffer. Then, SOMAmer-protein complexes are captured on a second set of streptavidin-coated beads (Catch-2), SOMAmers are released from proteins using a denaturating buffer (Catch-3), and finally hybridized to complementary sequences on microarrays for quantification by fluorescence [23].

Spiegelmers, modified RNA aptamers in an L-configuration (Spiegel = mirror), have higher stability and affinity as compared with standard RNA aptamers; in particular, spiegelmers are more resistant to nuclease degradation and can hybridize with nucleic acids at very low rates [3]. Therefore, spiegelmers can have a longer half-life in body fluids with better pharmacokinetics and pharmacodynamics (PK/PD) properties as compared with standard aptamers, thus, making these molecules more suitable for drug delivery or as anticancer agents. Spiegelmers are selected through SELEX using the synthetic mirror image of the natural target for positive selection. Bound aptamers are in the natural D-configuration and are later amplified, cloned, and sequenced by stereoselective enzymes. Finally, the produced aptamer sequences are employed for synthetization of L-aptamers using enantiomeric (L-)ribonucleotides [27].

Finally, xenobiotic nucleic acid (XNA) libraries are the most recent approach to markedly increase aptamer stability [28]. XNAs are artificial and are different from natural DNA and RNA because of the presence of alternative backbone or sugar congeners making XNAs highly resistant to nuclease degradation [28,29,30]. Then, XNA aptamers are selected through the cross-chemistry X-SELEX approach using immobilized targets on solid phase or gel and, subsequently, the selected XNA aptamers are reverse transcript, amplified, or sequenced [30]. To date, XNA aptamers are available for targeting hen-egg lysozyme, human immunodeficiency virus (HIV) transactivating response RNA element and HIV-reverse transcriptase, human thrombin, human vascular endothelial growth factor 165, and human neutrophil elastase [28,29,30]. However, in vitro XNA selection is more challenging as compared with other aptamers [29].

## 3. Leukemias

### 3.1. Acute Lymphoblastic Leukemia

Acute lymphoblastic leukemia (ALL) is a malignant aggressive clonal hematological disorder of either a T or B cell origin and mostly affecting children between two and four years old and adults of age >50 years [1]. Patients usually present with lymphocytosis and lymphoblast infiltration of the BM; however, extramedullary infiltration is also common, such as the central nervous system (CNS), lymph node, and mediastinal involvement especially during T-ALL [1,2]. Some clinical entities show recurrent genetic abnormalities, such as translocation (9;22) involving *BCR* and *ABL1* genes, or t(12;21) resulting in the fusion protein translocation-Ets-leukemia virus 6(ETV6)- Runt-related transcription factor 1(RUNX1) [1]. Current treatments are affective in most of the cases utilizing a five-step therapeutic protocol (induction, consolidation, CNS sterilization, intensification, and maintenance) resulting in a five-year overall survival of 89% in those aged 20 or younger [31]. However, T-ALL, high and very-high risk B-ALL patients, especially those aged 35 or older, have a five-year overall survival of less than 40%, and relapse is registered in one-fifth of the cases [32]. Therefore, new therapeutic and molecularly targeted approaches are required for a more efficient treatment of ALL; in addition, more sensitive methodologies are essential for determination of the minimal residual disease (MRD) which is highly correlated to treatment responsiveness, disease relapse, and progression [33].

The aptamer Sgc8 was generated in 2006 by a cell-based SELEX where an ssDNA library of 52-mer random sequence regions was incubated with CCRF-CEM cells, an in vitro T-ALL model, and bound sequences were first eluted by heating and subsequently incubated with a B cell lymphoma cell line for counterselection, and later amplified. After a mean of 20 rounds of selection, Shanguann et al. identified two aptamers, sga16 and the homologues sgc8, which specifically bound the protein tyrosine kinase-7 (PTK-7) on CCRF-CEM cells with high affinity (K_d_ = 0.8 nM) [34]. Therefore, this aptamer can be efficiently used for cancer cell detection when conjugated with fluorescent probes and employed for MRD determination [35]. Sgc8 has already been conjugated to heavy metals, fluorescent silica nanoparticles, and magnetic beads [35,36,37]. In particular, terbium ion (Tb^3+^), a rare earth element with long fluorescence lifetime and narrow emission spectrum that is enhanced by ssDNA binding, can be coupled with Sgc8, and Tb^3+^-aptamer complexes can be used for quantification of CCRF-CEM cells by fluorescence with a lower detection limit of 5 cells/mL and linearity in a wide range of cell concentrations (5–5 × 10^6^ cells/mL, R^2^ = 0.9881) without affecting cell viability [36,38,39,40,41]. This method has the highest sensitivity and specificity for cancer cell detection as compared with other published procedures [38,39,40,41]. Scg8 conjugated with magnetic beads and a rolling cycle amplification probe can be employed for efficient MRD measurement with a sensitivity of one cell in 20,000, much lower than the accepted threshold of 0.01% for T-ALL or one cell in 10,000 [35]. In addition, amine-labeled Sgc8 can be combined with carboxyl-modified fluorescent silica nanoparticles (FSNPs-COOH) containing a water-in-oil microemulsion of fluorescein isothiocyanate (FITC), 3-aminopropylmethyldimethoxysilane (APTMS), cyclohexane, *n*-hexanol, Triton X-100, distilled water, tetraethyl orthosilicate (TEOS), ammonium hydroxide, and NTTS 31. These Sgc8-labeled FSNPs can efficiently and specifically identify CCRF-CEM cells by flow cytometry and fluorescence microscopy without affecting cell viability [37].

Nanoparticles are widely used for a more efficient drug delivery in the tumor site as these systems have prolonged circulatory half-life and show increased uptake in neoplastic cells, thus, targeting tumors in a more specific manner as compared with standard drug formulations [4,5]. Therefore, aptamers can be conjugated to nanoparticles increasing the uptake of chemotherapeutic agents in tumor cells. Sgc8 has been incorporated in gold nanoparticles alone or in combination with doxorubicin or daunorubicin showing an increased cytotoxic effect against cancer cells [5,42,43]. In addition, a self-assembling glutathione-responsive prodrug based on Sgc8 and cytarabine (Ara-C) has been designed and tested in vitro and in a CCRF-CEM tumor-bearing mouse model showing increased tumor growth inhibition and survival rate and reduced side effects as compared with standard Ara-C formulation [44]. However, to date, the only registered clinical trial is an early phase I study for assessment of safety and pharmacokinetics/pharmacodynamics in healthy volunteers for further application in treatment of colorectal cancers (Clinicaltrial.gov identifier, NCT03385148).

CD20 which is a non-glycosylated tetra-span membrane antigen-like phosphoprotein encoded by the *MS4A* gene is expressed on B cells from late pro-B to the memory stage and functions as Ca^2+^ channel activated by B cell antigen receptor (BCR) induction. CD20 could trigger the activation of several Src tyrosine kinases, such as Lyn and Lck, involved in B cell proliferation and differentiation [45,46]. This protein is variously expressed on B cell neoplastic clones in malignant hematological diseases, such as B-ALL, CLL, or non-Hodgkin’s lymphomas, and also on B cells during autoimmune disorders and non-malignant hematological diseases, such as idiopathic thrombocytopenic purpura. Thus, CD20-targeting therapeutic approaches, mostly monoclonal antibodies, have dramatically changed the outcomes of patients with autoimmune and hematological disorders [47]. The AP-1 aptamer has been screened by cell-SELEX using a CD20-transfected human embryonic kidney cell line (HEK293T), showing a high binding affinity for the CD20 antigen (K_d_ = 96.91 ± 4.5 nM) and high specificity and sensibility for recognition by flow cytometry of CD20^+^ blasts in BM specimens obtained from B- and T-ALL patients [48]. Therefore, aptamers targeting surface markers can be used for tumor cell detection as chemical antibodies conjugated to standard fluorochromes, and for receptor-mediated internalization for drug delivery.

### 3.2. Acute Myeloid Leukemia

AML is a heterogeneous group of clonal aggressive hematologic malignancies characterized by a block of differentiation and increased proliferation of myeloid neoplastic cells harboring various cytogenetic abnormalities [1,2,49]. Patients are usually treated with a two-step therapeutic protocol as follows: a first induction phase using a seven-day Ara-C administration and a three-day anthracycline (daunorubicin or idarubicin) therapy; and a second consolidation phase if complete remission is achieved with post-remission therapy or hematopoietic stem cell transplantation (HSCT) when a donor is available [1]. However, 10%–40% of patients are refractory and two-thirds relapse with poor outcomes especially in elderly unfit for intensive chemotherapy showing a 10-month overall survival [50]. Therefore, new therapeutic approaches are required to increase response rates in AML patients and to reduce adverse drug-related effects. Aptamers can be used to selectively target highly expressed proteins in cancer cells and to deliver chemotherapeutic agents in a tumor-site specific manner [4,5].

AS1411 which is a synthetic 26-mer phosphodiester oligodeoxynucleotide and longer versions (e.g., GRO29A) is a nucleolin-targeting aptamer forming various G-quadruplex-containing structures such as monomolecular quadruplexes or different mixtures of monomer and dimer quadruplexes [51]. Nucleolin, a nucleolar non-ribosomal phosphoprotein, is involved in several intracellular signaling pathways, such as cell growth and proliferation or regulation of rRNA transcription and is abnormally expressed and localized in highly proliferating cancer cells [52]. Nucleolin is also downstream of several cytokine- and growth factor-mediated signaling pathways including tumor growth factor(TGF)-β, C-X-C Motif Chemokine Receptor(CXCR)4, C-C chemokine receptor(CCR)6, or epidermal growth factor, and downstream of NFκB (nuclear factor kappa-light-chain-enhancer of activated B cells) and DNA methyltransferases [4]. In addition, nucleolin can bind the AU-rich element of BCL2 mRNA and stabilize its translation exerting anti-apoptotic effects [5]. On the basis of this evidence, AS1411 has been first tested in vitro and in vivo xenograft models to study cytotoxic effects of this aptamer on cancer cells [53,54,55,56]. AS1411 which displayed growth inhibitory activities blocking the S phase of cell cycle in prostate, breast, HeLa, leukemia, and lymphoma cancer cell lines, and biodistribution and pharmacokinetics were investigated in mice bearing lung and renal human tumor xenografts injected with a single or multiple intravenous bolus of [^3^H]AS1411 (1, 10, and 25 mg/kg) or as continuous infusion for four days. Antitumor activity has also been investigated in vivo in nude mice bearing human tumor xenografts derived from A549 non-small cell lung cancer cells, from A498 renal cancer, SKMES lung cancer, or MX1 breast cancer cells showing a delay of tumor growth in a dose-dependent manner and enhanced when combined with gemcitabine (160 mg/kg) in mice bearing human xenografts derived from PANC-1 pancreatic cancer cells [56]. In a phase I clinical trial, AS1411 was safe and well tolerated in rats and dogs. To date, only two clinical trials have been completed for evaluation of AS1411 efficacy in combination with Ara-C for the treatment of primary refractory or relapsed AML (Clinicaltrials.gov identifiers, NCT01034410 and NCT00512083) (Table 2) showing positive preliminary results and synergistic anticancer activity with Ara-C [56].

CD33, also known as Siglec-3 as a member of the sialic acid-binding immunoglobulin-like lectins, is a 67 kDa transmembrane glycoprotein with an extracellular domain characterized by a sialic acid immunoglobulin(Ig)-like binding domain and a C2 Ig-like domain, and a cytoplasmic tail with an immunoreceptor tyrosine based inhibitory motif (ITIM) [57]. Siglec-3/CD33 mainly acts by suppressing the expression of pro-inflammatory cytokines, such as tumor necrosis factor(TNF)-α [57,58]. CD33 is prevalently expressed on normal multipotent myeloid precursors and circulating monocytes, dendritic cells, and some subsets of B and T cells, while mature neutrophils and macrophages have low surface levels of this receptor [58]. Conversely, CD33 is present on AML cells especially in acute promyelocytic leukemia and in blasts harboring *nucleophosmin1 (NPM1)* or FLT3 internal tandem duplication (*FLT3/ITD*) mutations, and patients with increased CD33 levels experience shorter overall survival [59]. A 25-mer DNA aptamer binds to CD33 and can be internalized by CD33^+^ myeloid cells with a K_d_ of 17.3 nM [60]. In addition, a dual aptamer targeting CD33 and CD34 has been screened by cell-SELEX process, and can be used for leukemic cell detection or drug delivery using gold functionalized nanoparticles containing AML-M2 overexpressed gene-targeting ASOs, such as *Bcl-2* or *XIAP* [61]. However, no clinical trials have evaluated the efficacy and safety of these aptamers in AML treatment.

CD117, also known as c-KIT, is a stem cell marker and is required for hematopoietic stem cell (HSC) maintenance in the BM and for extramedullary hematopoiesis [62], and upon stimulation by the stem cell factor, c-KIT phosphorylates and activates several signaling pathways, such as JAK/STAT or phosphoinositide 3-kinase (PI3K), leading to cell survival, proliferation, and differentiation [62]. CD117 is highly expressed in many solid cancers and leukemias, thus, making this marker suitable for molecular targeting [63]. A CD117-targeting aptamer was developed by cell-SELEX using HEL cells, a CD117-expressing cell line, and subsequently coupled to activated methotrexate using a 5′-amino-linked oligonucleotide 5AmMC6 as linker. This aptamer-drug complex is highly internalized by CD117^+^ leukemic cells causing G1 phase arrest and increased apoptosis in both cell lines and primary BM cells from AML patients with minimal toxicity on non-leukemic cells [64].

CD123 is the interleukin(IL)-3 receptor and upon IL-3 stimulation, triggers several signaling pathways, such as JAK/STAT, leading to cell survival and proliferation, and is normally present on hematopoietic progenitors of both myeloid and lymphoid lineages [65]. CD123 is overexpressed in both leukemia and lymphoma disorders and in both leukemic stem cells and more differentiated blasts. Two 66-mer aptamers, ZW25 and CY30, have been selected by SELEX against an extracellular CD123 epitope and the ZW25 was subsequently used to create a self-assembled targeted drug train (TDT) using probes rich in C/G in which doxorubicin can intercalate. These aptamer-TDT complexes are selectively internalized by the endosome/lysosome pathway in leukemic cells and doxorubicin is released into the cytoplasm ultimately interfering with DNA replication and blocking tumor cell growth in both in vitro and in vivo xenograft model developed by injecting Molm-13 cells in BALB/c mice [66].

### 3.3. Chronic Myeloid Leukemia

CML is a rare clonal hematological condition characterized by an early indolent slow-growing phase, an accelerated phase, and a late blastic phase with signs and symptoms of acute leukemia [1,2]. Blood film displays immature forms of neutrophils including myeloblasts, myelocytes, metamyelocytes, and band forms having a common chromosomal abnormality known as “Philadelphia chromosome” deriving from a reciprocal translocation t(9;22)(q34;q11) [1]. The fusion protein BCR-ABL and its mutant forms can activate several signaling pathways, such as GRB2, STAT5, PI3K, phospholipase Cγ, protein kinase B (AKT), or Src family kinases, ultimately leading to activation of transcription factors including NF-κB and MYC involved in expression of genes related to cell survival and proliferation. In addition, BCR-ABL can also increase the expression of several pro-inflammatory cytokines and growth factors [67]. β-arrestins which are scaffolding proteins for regulation of signal transduction at G protein-coupled receptors are essential for the activation of the Hedgehog/Smoothened and Wingless/Frizzled signaling pathways which are crucial for normal hematopoiesis and seem to be involved in CML pathogenesis [3].

β-arr2A3 which is an 80-mer RNA aptamer, has been screened by SELEX together with two other aptamers, β-arr2A1 and β-arr2A2, with various binding affinity (β-arr2A1, K_d_ = 19.83 nM; β-arr2A2, K_d_ = 4.13 nM; and β-arr2A3, K_d_ = 22.03 nM) and high selectivity for β-arrestin (β-arr2A1, K_d_ = 965.5 nM; β-arr2A2, K_d_ = 2159.0 nM; and β-arr2A3, K_d_ = 748.5 nM). Functionalized aptamers as nucleolin aptamer chimeras are efficiently internalized and can reduce signaling transduction in the K562 cell line, an in vitro CML model, and lymphoblastoid cells, a type of non-cancerous human B cells, also decreasing leukemic cell growth [68]. However, no ongoing clinical trials are registered.

### 3.4. Chronic Lymphocytic Leukemia

CLL, which is the most common type of leukemia in Western countries, is characterized by the presence of CD5^+^ clonal B lymphocytes in peripheral blood, lymph nodes, BM, and spleen. CLL patients show different clinical courses as the five-year overall survival ranges from 93.2% to 23.3% [69,70,71]. This clinical heterogeneity mirrors diverse molecular signatures with various prognostic values, such as the mutational status of immunoglobulin variable region (*IGHV*) gene, CD38 surface expression [72,73], and other chromosomal alterations [71,74]. In particular, the TP53 status (no abnormalities vs. del [17p] or *TP53* mutation or both) and the *IGHV* mutational status (mutated vs. unmutated) are considered major prognostic markers in CLL, and have been included in a prognostic index, the International Prognostic Index for Chronic Lymphocytic Leukemia (CLL-IPI), together with clinical features and serum markers [75]. Other emerging biomarkers are mutations in *SF3B1*, *ATM*, *NOTCH1*, *MYD88*, and *BIRC3* [71]. *TP53* or *BIRC3* (Baculoviral IAP Repeat Containing 3) abnormalities are linked to high risk of clonal evolution and transformation, treatment resistance, and poor prognosis; *NOTCH1* or *SF3B1* (splicing factor 3B subunit 1) mutations or del11q22-q23 to intermediate-risk; +12 or normal genetics to low risk; and del13q14 to a very-low risk of transformation [76]. In addition, CLL cell survival also relies on the crosstalk between neoplastic cells and microenvironment via soluble and cell-surface-bound factors, such as the CXC chemokine ligand(CXCL)12, CD40 ligand, or CD49d [77].

The CXCL12-CXCR4 axis is important for retention of leukemic cells in the BM, and BM stromal cells constitutively secrete CXCL12 in a decreasing oxygen-concentration gradient manner. CXCL12, then, binds glycosaminoglycans on the cell surface or in the extracellular matrix and the amino-terminal domain is exposed for binding to CXCR4 attracting CLL cells to the BM niche [78]. In addition, this CXCL12-CXCR4-mediated close contact between neoplastic cells and stromal cells has anti-apoptotic effects and induces cell growth and survival through the activation of several signaling pathways such as STAT3, AKT, and ERK1/2 [77]. NOX-A12 which is a 45-mer L-stereoisomer RNA spiegelmer was first developed by Noxxon Pharma (Germany) for targeting CXCL12 and disrupting the CXCL12/CXCR4 axis [3]. NOX-A12, or Olaptesed pegol, can inhibit the CXCL12-directed chemotaxis in a dose-dependent manner not only in primary CLL cells, but also in Jurkat T-cell line and Nalm-6 ALL cell line [77,79]. Outside the protective BM environment, CLL cells are more susceptible toward cytotoxic agent effects, such as when cells are cotreated with NOX-A12 and bendamustine or fludarabine [80,81,82]. These promising preclinical data have supported the approval of a phase I study (NCT00976378) for investigation of safety, tolerability, and PK/PD profiling of single intravenous NOX-A12 dose in healthy male and female subjects. A second phase I clinical trial (NCT01194934) has been conducted for assessing safety, tolerability, PK/PD, and mobilization rates of repeated intravenous doses (2.0 and 4.0 mg/kg/d) alone or in combination with filgrastim showing good safety and tolerability profiles [81]. Afterwards, NOX-A12 has been evaluated in combination with bendamustine and rituximab for treatment of relapsed/refractory CLL patients in a phase I/II clinical trial (ClinicalTrials.gov identifier, NCT01486797). A total of 28 patients was recruited, and three of them achieved a complete remission, and 21 a partial response, including high-risk patients with a 17p deletion, with more than 80% of patients alive at a 28-month follow-up. In addition, patients who have received previous chemotherapeutic agents (one or more therapeutic lines) also achieved an overall response rate >80% [82].

Another emerging candidate molecular target for CLL treatment is the elongation factor 1 A (eEF1A) involved in the enzymatic delivery of aminoacyl-tRNAs to the ribosome (“canonical function”) and in cell cycle and apoptosis regulation (“non-canonical functions”) [83,84]. The ubiquitous isoform eEF1A1 is increased in CLL lymphocytes as compared with healthy controls, and high levels are related to poor outcomes in patients [84]. The aptamer GT75, a 75-mer GT repetition containing oligonucleotide, can efficiently bind eEF1A1 and significantly reduce viability and tumor growth in several in vitro cancer models, such as human B-CLL MEC-1 cell line, T-lymphoblastic leukemia cell line (CCRF-CEM), and hepatocellular carcinoma cell line [83,84,85].

## 4. Lymphomas

### 4.1. Hodgkin Lymphoma

Hodgkin lymphoma (HL) is a clonal hematological disorder derived from the neoplastic transformation of a germinal center B cell, the so-called Hodgkin and Reed-Sternberg (HRS) cells, surrounded and supported by an inflammatory infiltrate (pabulum) of B and T lymphocytes, plasma cells, leucocytes, eosinophils, and mast cells. Therefore, neoplastic cells are rare within the tumor mass accounting for 1%–5% of the total cellularity, thus, making HL diagnosis difficult from peripheral blood specimens. The HRS cells display positivity for CD30 and CD15 surface markers, for p16^INK4a^ and p21^Cip1^ cell cycle inhibitors and senescence markers, and for the p65 subunit of NF-κB [86]. Therefore, CD30 is actually used for both detection of neoplastic cells and for treatment of advanced, relapsed, and refractory HL, or for preventing disease progression after transplantation in high-risk patients [87].

To date, available aptamers identify CD30 epitopes for identification of HRS helping diagnosis of HL [88,89]. A 39-mer RNA aptamer has been labeled with a Cy5 dye at the 5′-end and used for detection of CD30-expressing cell lines (SUDHL-1, Karpas 299, L340, HDLM2, L428, and KMH2 cells) and of CD30^+^ Karpas 299 cells spiked in healthy BM samples by flow cytometry [89]. Two ssDNA aptamers, PS1, and its truncated form PS1NP have been identified by cell-SELEX using HDLM2 and Karpas 299 cells for positive selection and U937 for counterselection [88]. These aptamers have a maximal cell binding affinity of 10 nM (PS1, K_d_ = 5 ± 0.5 nM; PS1NP, K_d_ = 10 ± 0.6 nM), and high stability in serum as 40%–50% of aptamers are still intact after 24 h incubation at 37 °C. In addition, Cy3-labeled aptamers can specifically identify HL cells from cell lines (SUDHL-1, Karpas 299, HDLM2, and KMH2 cells) and spiked in whole blood or U937 cells [88].

### 4.2. Anaplastic Large Cell Lymphoma

Anaplastic large cell lymphoma (ALCL) is a mature T cell neoplasm according to the 2016 revision of the World Health Organization classification of lymphoid malignancies [1,2]. Neoplastic cells always have CD30 positivity; however, the following two clinical entities are distinguished based on anaplastic lymphoma kinase (ALK) expression: ALK^+^ with a recurrent t(2;5)(p23;q35) leading to a fusion ALK-NPM1 protein; and ALK^−^ clinical form with two recurrent chromosomal abnormalities, such as t(6;7)(p25.3;q32.3) involving *Dual Specificity Phosphatase 22*(*DUSP22)-Interferon Regulatory Factor 4(IRF4)* and (fragile site, aphidicolin type, common, fra(7)(q32.3) (*FRA7H)*, and rearrangements involving the *TP63* gene [1,90]. The ALK^+^ ALCL form presents more frequently during childhood as an advanced stage disease with adenopathy and BM infiltration, whereas ALK^−^ ALCL is an adult disease, mostly diagnosed in 40 years old adults [1]. Outcomes are also different as the five-year overall survival of ALK^+^ ALCL is 80% with standard chemotherapy as compared with a 20% five-year overall survival for the ALK^−^ form [91].

CD30 positivity of ALCL cells can be used for detection of neoplastic cells by flow cytometry using dye-labeled aptamers, as described above, or for cell-specific drug delivery [92,93]. A polyethyleneimine (PEI)-citrate nanocore can be loaded with a synthetic RNA-based CD30 aptamers and ALK siRNA and used for selective uptake of siRNAs by CD30^+^ cells and ALK silencing resulting in decreased expression of the NPM-ALK fusion protein in Karpas 299 cell line. In addition, these nanocomplexes can induce tumor growth inhibition and apoptosis [92]. Synthetic CD30 aptamers can be also loaded on hollow gold nanosphere and used for selective doxorubicin released into cells in a pH-dependent manner [93]. Indeed, these nanospheres efficiently and specifically bind both HL cell (HDLM2, KMH2, L428, and L540) and ALCL cell lines (SUDHL-1 and Karpas 299), and, once internalized into lysosomes, they rapidly release most of their doxorubicin cargo under a pH of 5.0, while being stable in physiological conditions (pH 7.4) [93]. Finally, synthetic ssDNA CD30 aptamer, ALK siRNA, and doxorubicin can be self-assembled together forming a multifunctional aptamer nanomedicine (Apt-NMed). In this system, the CD30 aptamer allows selective uptake of Apt-NMed by CD30^+^ cells, while ALK siRNA gene silencing and doxorubicin direct cytotoxic effect in both in vitro and in vivo xenograft mouse models [94].

### 4.3. Burkitt Lymphoma

Burkitt lymphoma (BL) which is a mature B cell neoplasm can present in the following two clinical entities: the endemic (African) form is caused by Epstein–Barr virus (EBV) infection with a specific translocation t(8;14)(q24;q32) involving *MYC/c-Myc* and the heavy chain immunoglobulin genes; and a sporadic form accounting for 1%–2% of adult lymphomas and for about 40% of pediatric diseases [1,2].

TD05 which is a 48-mer aptamer or its 44-mer truncated form has been selected by cell-SELEX using the Ramos cell line, an EBV^−^ BL cell line derived from a three-years-old Caucasian male, and selectively binds to the immunoglobin heavy mu chain (IGHM) which is the heavy chain IgM region [3,4,79]. TD05 can be conjugated to a photosensitizer, such as Ce6, an oxidative stress inducer upon irradiation leading to increased cell death in target cells as compared with the control cells (CEM, K562, NB4, and HL60 cell lines) [95]. TD05 can also be attached to a lipid tail forming a self-assembling nanostructure for a faster antigen recognition and higher dynamic binding affinity [96]. Moreover, TD05 can be conjugated to a cyanin(Cy)5 dye or a deoxyribonuclease (DNase)-activatable fluorescence probe for specific recognition of the Ramos cells by flow cytometry or in vivo fluorescence imaging [97,98].

### 4.4. Diffuse Large B Cell Lymphoma

Diffuse large B cell lymphoma (DLBCL) is a mature B cell neoplasm divided into the following four molecular subtypes: germinal center B-cell-like (GCB), activated B cell-like (ABC), primary mediastinal large B cell lymphoma (PMBCL), and unclassified. The GCB type is characterized by increased expression of BCL6 and CD10, the absence of IRF4 and PR domain zinc finger protein 1 (BLIMP1), and highly mutated Ig genes; whereas the ABC subtype has a molecular profile similar to that of BCR-activated B cells or plasma blasts [99]. Genetic lesions common to both GCB and ABC are the following: oncogenic activation of *BCL6*; alterations of histone modification genes, such as *CREB binding protein* (*CREBBP)*; and inactivating mutations of *TP53*, *β2 microglobulin* (*B2M)*, or *CD58* [99,100,101]. Chromosomal translocations of *BCL2* and *MYC* are typical of the GCB type, as well as mutations in the polycomb-group oncogene *EZH2* (Enhancer of Zeste 2 Polycomb Repressive Complex 2 Subunit) or in proteins of the G-protein coupled inhibitory circuit for growth and localization of B cells in the germinal center [99]. Conversely, the ABC subgroup is characterized by genetic alterations causing constitutive activation of NF-κB, BCR, and Toll-like receptor (TLR) signaling pathways [99,101]. The B cell-activating factor (BAFF), a trimeric-surface or soluble factor essential for B cell maturation, is produced by neutrophils, monocytes, and dendritic cells, and can bind the following three different receptors: B cell maturation antigen (BCMA); transmembrane activator and calcium modulating ligand (CAML) interactor (TACI); and the BAFF-receptor (BAFF-R), crucial for B cell survival and maturation. BAFF-R is usually expressed in the mantle zone, germinal center, and scattered in the interfollicular area [102]. According to the proposed origin of DLBCL cells, GCB lymphomas are frequently positive for BAFF-R and increased expression is related to a slow tumor growth [102]. Indeed, patients with negative BAFF-R expression present with a more severe disease and experience a shorter progression-free survival and overall survival as compared with those with BAFF-R expression at diagnosis [103]. Several 2′-fluoropyrimidine modified RNA aptamers have been identified after 10 rounds of SELEX for binding BAFF-R with high affinity as follows: R-1, K_d_ = 47.1 ± 7.6 nM; R-2, K_d_ = 95.3 ± 20.2 nM; and R-14, K_d_ = 95.7 ± 18.7 nM. These aptamers are efficiently internalized by BAFF-R-expressing Jeko-1 cells, a peripheral blood mantle cell lymphoma cell line, but not by CCRF-CEM cells, a T-cell lymphoblast-like cell line, and can reduce proliferation rates in a dose-dependent manner of various B cell lymphoma cell lines according to their initial BAFF-R expression [104]. Aptamers can be linked to siRNAs forming chimeras for *STAT3* gene silencing or can be conjugated to siRNAs using a ”stick” sequence of 16 nucleotides appended to the aptamer 3′-end, and a linker of seven three-carbons to avoid steric interactions [104].

In addition, the anti-nucleolin AS1411 aptamer can be used in DLBCL because it can act synergistically with doxorubicin by interfering with the DNA repair machinery, such as the topoisomerase-II-alpha [4].

## 5. Multiple Myeloma

Multiple myeloma (MM) is a clonal hematological disorder characterized by the expansion of malignant plasma cells producing monoclonal paraprotein (M protein) in the BM and subsequent bone destruction and displacement of normal HSCs [105]. Evidence shows that neoplastic plasma cells could derive from long-lived post-germinal center plasma cells; however, genetic alterations, such as IgH translocations or hyperdiploidy, likely occur during the isotype class switching and somatic hypermutation leading to increased genomic instability with de novo mutations and tumor heterogeneity [106,107]. The IgH translocation could involve three gene families, i.e., cyclin d (*CCND*), *MAF*, and Wolf–Hirschhorn syndrome candidate 1/Fibroblast growth factor receptor 3 (*MMSET/FGFR3*). These genetic abnormalities differently affect gene expression in MM cells influencing the interaction between the neoplastic clone and BM microenvironment [106]. Several epigenetic alterations, including non-coding RNA expression, could also play an important role in sustaining survival of neoplastic clones [105,108], and increased MYC expression, activating *K-RAS* mutations or chromosome 13 deletion can additionally accelerate the transition from the asymptomatic premalignant form to MM [106]. Therefore, MM is an heterogenous clinical entity with various distinct cytogenetic abnormalities and clinical features which influence therapeutic strategy choice currently based on tumor stage and disease biology [109,110].

CD38 which is a 45 kDa surface glycoprotein is an activation marker, an adhesion molecule, and an ectoenzyme for the metabolism of extracellular NAD^+^ and intracellular NADP. In physiological conditions, CD38 is expressed at low levels on myeloid and lymphoid cells, while present at high levels in neoplastic cells, thus, making this molecule a good therapeutic target in MM [111,112]. CD38-targeting ssDNA aptamers have been identified by cell-SELEX using MM1S cell line and CD38^−^ HDLM2 cells for counterselection, and subsequently conjugated to doxorubicin which exerts the antitumor activity after a pH-dependent release into lysosomes [113]. This aptamer-doxorubicin system can specifically kill MM cells and reduce tumor growth [114].

Annexin A2 (ANXA2), which is a member of the annexin family with Ca^2+^-mediated phospholipid-binding properties, is highly expressed on MM cells and could sustain cell adhesion and growth of the neoplastic clone in the BM niche [115]. The wh6 ssDNA aptamer has been selected using a nine-round SELEX for selectively binding ANXA2 (K_d_ = 8.75 ± 1.26 nM) both in vitro and in vivo xenograft model. The wh6 aptamer can also be conjugated to a fluorochrome, such as the FITC, and used for MM cell detection by flow cytometry for diagnostic purposes [116].

## 6. Graft versus Host Disease

Graft-versus-host disease (GvHD) is an acute or chronic life-threating condition developed after allogeneic HSCT and caused by the following: (i) the presence of immunologically competent mature T cell population in the graft, (ii) the host or recipient is not able to reject transplanted cells, and (iii) recipient tissue antigens are different from those present in the donor. The activation of antigen-presenting cells (APCs) can initiate GvHD and cells can be primed by conditioning regimens that damage tissues and release pro-inflammatory cytokines, such as IL-1 and TNF-α, from activated macrophages [117]. Activated CD8^+^ T cells mostly sustain tissue damage; however, CD4^+^ T cells also play an important role in GvHD initiation and maintenance [117,118]. Indeed, severity and time of onset of GvHD seem to be related to various proportions of T regulatory, T helper(Th)1, Th2, or Th17 cells, the latter are activated also by plasmacytoid and myeloid dendritic cells [117,119]. In addition, several pro-inflammatory cytokines are involved in GvHD pathogenesis, such TNF-α and Th17-related cytokines including IL-6, IL-1β, IL-17, IL-21, IL-23, and IL-23R [117]. On the basis of the major role of the immune system in GvHD pathogenesis, aptamers developed for modulating immune cells and response in autoimmune disorders can have a rationale for testing in prevention and treatment of this post-transplant complication. For a more detailed description, please refer to Oelkrug et al. [118]. However, two 2′-fluoropyrimidine modified RNA aptamers, CD28Apt2 and CD28Apt7, against the CD28, a costimulatory molecule involved in rapid T cell activation, can be used for modulating T cell responses as they can be easily engineered to act as CD28 agonists or antagonists [3,120].

## 7. Aplastic Anemia

Acquired aplastic anemia (AA) is a BM failure syndrome characterized by pancytopenia and BM hypocellularity likely caused by an autologous immune attack against hematopoietic stem and progenitor cells [121,122]. Hematologic improvement of blood counts after immunosuppressive therapies (IST) is one of the most supportive indirect evidence of autoimmunity in AA [123]. Additional lines of indirect evidence are increased activated cytotoxic T cells that inhibit BM proliferation, oligoclonality of effector memory T cells, TNF-α-producing macrophages, and the presence of pro-inflammatory cytokines [121,122,123,124,125,126]. The SOMAscan assay has been used for large-scale proteomics profiling of serum and plasma of AA patients at diagnosis and after IST with or without thrombopoietin receptor agonist eltrombopag [127]. More than 600 proteins were found different in serum or plasma of AA patients as compared with controls and proposed for further validation. In particular, 19 serum proteins were identified as candidate biomarkers of responsiveness to IST and 15 out of these were also higher in the plasma of AA patients as compared with healthy controls. Among those proteins, four (Dickkopf WNT signaling pathway inhibitor 1 [DKK1], L-selectin, CCL17, and hepatocyte growth factor [HGF]) were successfully validated by ELISA (enzyme-linked immunosorbent assay) confirming the high sensibility and specificity of the SOMAscan assay, thus, proposing four new biomarkers for diagnosis of AA and two (DKK1 and CCL17) as candidate predictors of responsiveness to standard IST. In addition, large-scale proteomics profiling can identify not only differentially present proteins between patients and healthy controls but can also relate those proteins to novel protein pathways, thus, broadening current knowledge on pathophysiology of studied diseases and opening new scenarios for discovery of novel molecular therapeutic targets [127].

## 8. Sickle Cell Disease

Sickle cell disease (SCD) is benign hematological diseases caused by a point mutation in the beta globin gene leading to a single amino acid change (Glu6Val) in the translated protein producing the assembling of the hemoglobin S (HbS) in the homozygous or double heterozygous condition [128]. In the deoxygenated conformation, the HbS is more susceptible of an intracellular 14-strand polymerization through a two-step process termed “double nucleation” [129]. This formation of HbS fibers in the cytoplasm leads to more rigid and sickle-shaped red blood cells (RBCs) which are not more able to easily pass through capillaries causing sub occlusions or real occlusions with ischemic events (e.g., stroke or priapism) [130]. The interaction between the αVβ3 integrin and P-selectin on activated endothelial cells seems to play a crucial role in the flow adhesion of RBCs to the endothelium and subsequent vaso-occlusion [131]. Several aptamers have been developed for disrupting the endothelial cell–RBC interaction: αVβ3 integrin-targeting aptamer; a PF377 aptamer binding P-selectin; and an anti-mouse 33-mer ARC5690 aptamer against P-selectin [129,130,131]. In addition, two noncompeting RNA aptamers, DE3A and OX3B, have been developed by SELEX alternating positive selection rounds with deoxy-HbS and negative counterselection with oxy-HbS [129]. These aptamers have high affinity to both deoxy- and oxy-hemoglobin S as follows: DE3A, K_d_ = 1.68 and 1.74 mM, deoxy-HbS and oxy-HbS, respectively; and OX3B, K_d_ = 8.57 and 3.56 mM, deoxy-HbS and oxy-HbS, respectively. In addition, both aptamers increase the delay time of polymerization in a dose-dependent manner, and DE3A causes a higher maximal level of inhibition of the polymerization rate, whereas OX3B seems to be more effective [129]. However, no ongoing clinical trials are evaluating the efficacy of those agents in SCD treatment.

## 9. Conclusions

Aptamers, which are short RNA or DNA oligonucleotides, specifically recognize proteins and small molecules based on their tertiary or quaternary structures acting as chemical antibodies. Although aptamers can be easily produced in vitro by SELEX or cell-SELEX and are less immunogenic as compared with monoclonal antibodies, we are far from using these molecules in routine clinical practice as anticancer agents or drug delivery system. In particular, unmodified aptamers are rapidly degraded in vivo by circulating extracellular endo- and exonucleases, thus, quickly cleared from the bloodstream by the kidneys. In addition, because nucleic acids are charged molecules and passive transport across cell membrane is difficult, it is challenging to reach a therapeutic aptamer concentration in the cytoplasm of target cells. Indeed, aptamers selected through cell-SELEX are internalized in a receptor-mediated endocytosis manner; however, in endosomes and lysosomes, oligonucleotides could also be degraded by nucleases, thus, only a small number of aptamers can reach the cytosol and exert the therapeutic action. Therefore, despite their therapeutic potential, only a few aptamers (e.g., AS1411 and NOX-A12) are under investigation for treatment of hematological disorders, such as AML or CLL, in combination with standard chemotherapeutic protocols. Conversely, aptamers can have a great potential in multiplex high-throughput proteomics analysis for screening of new biomarkers of diagnosis, prognosis, and responsiveness to treatment, thus, identifying new molecular therapeutic targets, as already reported for AA. In addition, because of their high affinity and selectivity to target molecules, aptamers can be efficiently employed for tumor cell detection by flow cytometry or other methodologies and can be used for more accurate MRD monitoring and also for diagnosis of malignant hematological diseases. In conclusion, we still have a long way to go before aptamers can be used as anticancer agents or drug delivery systems; however, in the meantime, we can largely explore their utility in clinical diagnosis and prognosis.

## Figures and Tables

**Figure 1 ijms-21-03252-f001:**
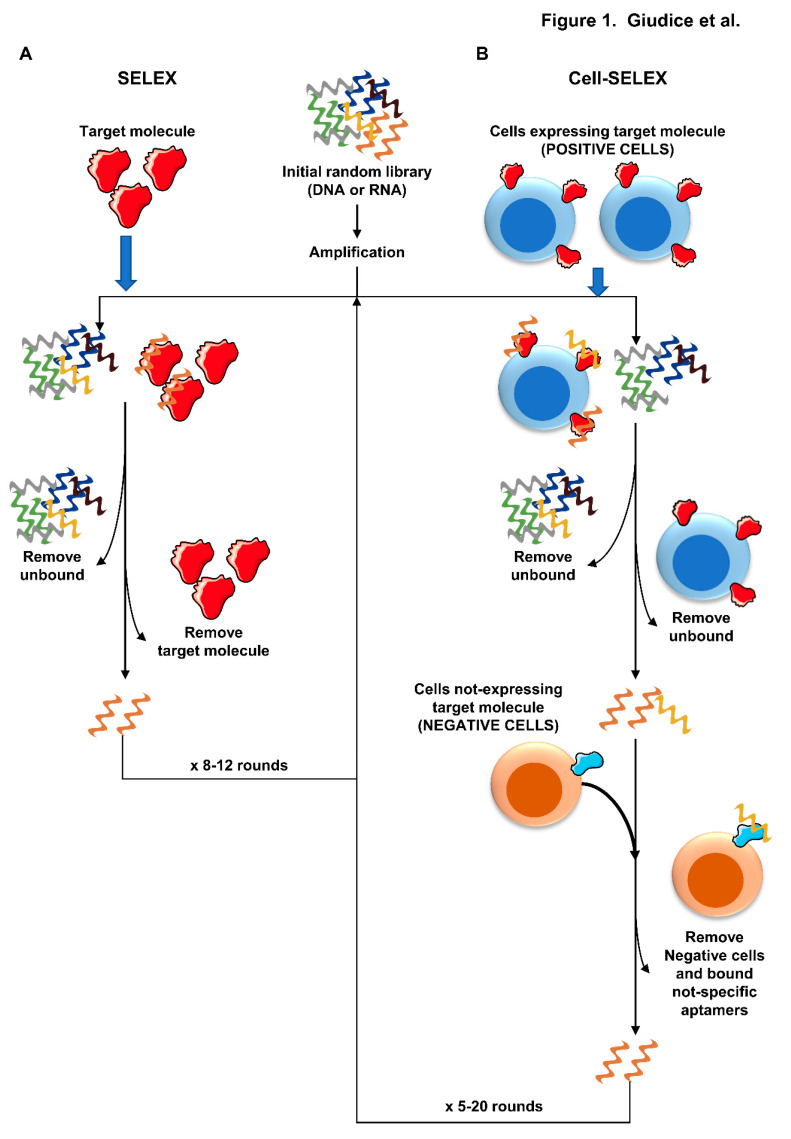
Schematic representation of (**A**) systemic evolution of ligands by exponential enrichement (SELEX) and (**B**) cell-SELEX processes (Some artworks from https://smart.servier.com/).

**Table 1 ijms-21-03252-t001:** Therapeutic antisense oligonucleotides (ASOs) for hematological diseases.

Name	Target	Indications	Combination Therapy	Clinical Trial	Ref.	Status
Cenersen	p53	MDS	Dexamethasone	Phase 1	NCT02243124	
BP1001	Grb2	Ph^+^ CML Ph^+^ AML High-risk Ph^+^ MDS	Dasatinib	Phase Ib/IIa	NCT02923986	Recruiting
		Recurrent Adult AML ALL MDS Ph^+^ CML	Low-dose Ara-C	Phase I	NCT01159028	Active, not recruiting
		AML High-risk MDS	Decitabine	Phase II	NCT02781883	Recruiting
BP1002	L-Bcl-2	Advanced B cell NHL		Phase 1	NCT04072458	Not yet recruiting
Oblimersen	Bcl-2	Recurrent B cell NHL	Rituximab	Phase II	NCT00054639	Completed
		Relapsed or refractory MM	Dexamethasone	Phase III	NCT00017602	Completed
		Waldenström Macroglobulinemia		Phase I/II	NCT00062244	Completed
		CML Ph^+^ chronic phase CML Relapsing CML	Imatinib	Phase II	NCT00049192	Completed
		Advanced lymphomas	Gemcitabine	Phase I	NCT00060112	Terminated
		CLL		Phase I/II	NCT00021749	Completed
		CLL	Rituximab Fludarabine	Phase I/II	NCT00078234	Completed
		Relapsed or refractory CLL	Filgrastim Cyclophosphamide Fludarabine	Phase III	NCT00024440	Completed
		Contiguous Stage II adult DLBCL Noncontiguous Stage II adult DLBCL Stage III adult DLBCL Stage IV adult DLBCL	Rituximab Cyclophosphamide Doxorubicin Vincristine Prednisone	Phase II	NCT00080847	Terminated
		Stage II, Stage III, or Stage IV DLBCL	Rituximab Cyclophosphamide Doxorubicin Prednisone Vincristine	Phase I	NCT00070083	Completed
		Stage II, Stage III, or Stage IV FL	Rituximab	Phase II	NCT00301795	Terminated
		Newly diagnosed Stage I, Stage II, Stage III, or Stage IV DLBCL	Rituximab Cyclophosphamide Doxorubicin Vincristine Prednisone		NCT00736450	Terminated
		Recurrent adult DLBCL Recurrent Grade 3 FL Recurrent MCL	Rituximab Ifosfamide Carboplatin Etoposide Filgrastim Pegfilgrastim	Phase I/II	NCT00086944	Completed
SPC2996	Bcl-2	CLL		Phase I/II	NCT00285103	Completed
G4460	c-myb	CML in chronic or accelerated phase	Filgrastim Busulfan Cyclophosphamide Autologous SCT	Phase II	NCT00002592	Completed
AEG35156	XIAP	Relapsed or refractory CLL Indolent B cell NHL		Phase I/II	NCT00768339	Terminated
		AML	Idarubicin Cytarabine	Phase II	NCT01018069	Terminated
		Refractory/Relapsed AML		Phase I/II	NCT00363974	Completed
EZN-2968	HIF-1α	Advanced jymphomas		Phase I	NCT00466583	Completed
AZD9150	STAT3	DLBCL	MEDI4736 Tremelimumab	Phase I	NCT02549651	Completed
IONIS-STAT3Rx	STAT3	DLBCL or other advanced lymphomas		Phase I/II	NCT01563302	Completed

Abbreviations: ASO, antisense oligonucleotide; MDS, myelodysplastic syndromes; Grb2, growth factor receptor-bound protein 2; Ph, Philadelphia chromosome; CML, chronic myeloid leukemia; AML, acute myeloid leukemia; Ara-C, cytarabine; ALL, acute lymphoblastic leukemia; Bcl-2, B cell lymphoma 2; NHL, non-Hodgkin lymphoma; MM, multiple myeloma; CLL, chronic lymphocytic leukemia; DLBCL, diffuse large B cell lymphoma; FL, follicular lymphoma; MCL, mantle cell lymphoma; SCT, stem cell transplantation; XIAP, X-linked inhibitor of apoptosis protein; HIF-1a, hypoxia-inducible factor 1-alpha; STAT3, signal transducer and activator of transcription 3.

**Table 2 ijms-21-03252-t002:** Aptamers in preclinical studies.

Name	Target	Selection	Positive Target	Length (nt)	K_d_ (nM)	Conjugation	Disease	Application
Sgc8	PTK7	Cell-SELEX	CCRF-CEM	41	0.8	Terbium ion (Tb3+)	ALL	Tumor cell detection LOD, 5 cells/mL
						Magnetic beads + RCA probe	ALL	MRD quantification LOD, 1 on 20,000 cells
						FSNPs-COOH	ALL	Tumor cell detection
						Gold nanoparticles Ara-C	ALL	Drug delivery
AP-1	CD20	Cell-SELEX	Transfected HEK293T	88	96.91 ± 4.5	FITC	ALL	Tumor cell detection
AS1411	Nucleolin	Non-antisense synthesize	-	26			AML	Antitumor activity in combination with Ara-C
							DLBCL	Antitumor activity
Anti-CD33	CD33			25	17.3	Gold nanoparticles + five ASOs + anti-CD34 aptamer	AML	Tumor cell detection Drug delivery
Anti-CD117	CD117	Cell-SELEX	HEL			MTX using a 5AmMC6 linker	AML	Tumor cell detection Drug delivery
ZW25 CY30	CD123	SELEX	-	66		Self-assembled TDT + Doxorubicin	AML	Tumor cell detection Drug delivery
β-arr2A3	β-arrestin2	SELEX	-	80	22.03	Chimera	CML	Antitumor activity
NOX-A12	CXCL12	SELEX	-	45			CLL ALL	Antitumor activity
GT75	eEF1A1			75			CLL ALL	Antitumor activity
Anti-CD30	CD30	SELEX		39		Cy5	HD ALCL	Tumor cell detection
						PEI-citrate nanocore + ALK siRNA	HD ALCL	Drug delivery
						Hollow gold nanosphere + Doxorubicin	HD ALCL	Drug delivery
						ALK siRNA + Doxorubicin	HD ALCL	Drug delivery
PS1 PS1NP	CD30	Cell-SELEX	HDLM2 Karpas 299		5 ± 0.5 10 ± 0.6	Cy3	HD ALCL	Tumor cell detection
TD05	IGHM	Cell-SELEX	Ramos	48		Ce6 photosensitizer	BL	Antitumor activity
						Cy5 DNase-activatable fluorescence probe	BL	Tumor cell detection
R-1 R-2 R-14	BAFF-R	SELEX	-		47.1 ± 7.6 95.3 ± 20.2 95.7 ± 18.7	siRNAs forming chimera or stick	DLBCL	Antitumor activity
Anti-CD38	CD38	Cell-SELEX	MM1S			Doxorubicin	MM	Drug delivery
wh6	ANXA2	SELEX	-		8.75 ± 1.26	FITC	MM	Tumor cell detection
DE3A OX3B	HbS	SELEX	-		1.68 8.57		SCD	Antithrombotic activity

Abbreviations: PTK7, protein tyrosine kinase-7; SELEX, systemic evolution of ligands by exponential enrichment; RCA, rolling cycle amplification; ALL, acute lymphoblastic leukemia; LOD, lower limit of detection; MRD, minimal residual disease; FSNPs-COOH, carboxyl-modified fluorescent silica nanoparticles; Ara-C, cytarabine; FITC, fluorescein isothiocyanate; AML, acute myeloid leukemia; DLBCL, diffuse large B cell lymphoma; ASO, antisense oligonucleotide; MTX, methotrexate; TDT, targeted drug train; CML, chronic myeloid leukemia; CLL, chronic lymphocytic leukemia; eEF1A1, eukaryotic translation elongation factor 1 alpha 1; Cy, cyanine; HD, Hodgkin disease; ALCL, anaplastic large cell lymphoma; PEI, polyethyleneimine; ALK, anaplastic lymphoma kinase; IGHM, immunoglobin heavy mu chain; BL, Burkitt lymphoma; DNase, deoxyribonuclease; BAFF-R, B cell-activating factor receptor; MM, multiple myeloma; ANXA2, annexin A2; HbS, hemoglobin S; SCD, sickle cell disease.

**Table 3 ijms-21-03252-t003:** Therapeutic aptamers for hematological diseases.

Name	Target	Indications	Combination Therapy	Clinical Trial	Ref.	Status
AS1411	Nucleolin	Primary refractory or relapsed AML	Cytarabine	Phase II	NCT01034410	Terminated
		Primary refractory or relapsed AML	Cytarabine	Phase II – dose escalating	NCT00512083	Completed
NOX-A12	CXCL12	Autologous SCT		Phase I	NCT00976378	Completed
		Hematopoietic SCT	Filgrastim	Phase I	NCT01194934	Completed
		CLL	Bendamustine Rituximab	Phase II	NCT01486797	Completed
		MM	Bortezomib Dexamethasone	Phase II	NCT01521533	Completed
NOX-H94	Hepcidin	Anemia End stage renal disease		Phase I/II	NCT02079896	Completed
ARC1779	vWF	TTP Von Willebrand disease Type-2b		Phase II	NCT00632242	Completed
ARC19499	TFPI	Hemophilia		Phase I	NCT01191372	Terminated

Abbreviations: AML, acute myeloid leukemia; MDS, myelodysplastic syndromes; CXCL12, C-X-C motif chemokine 12; SCT, stem cell transplantation; CLL, chronic lymphocytic leukemia; MM, multiple myeloma; vWF, von Willebrand factor; TTP, thrombotic thrombocytopenic purpura; TFPI, tissue factor pathway inhibitor.

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
