# Peer review of "Aptamers and Antisense Oligonucleotides for Diagnosis and Treatment of Hematological Diseases"

_ijms, 2020, doi:10.3390/ijms21093252_

Round 1

Reviewer 1 Report

The Review discusses the current state of the art in therapeutic aptamers and antisense oligonucleotides in the diagnosis and treatment of hematological diseases.  The manuscript is well written and would be a useful source in the area in terms of describing the aptamers that have been developed in this area, the format and focus of the review need to be revised.  For instance, the discussion of the methods for the development of modified aptamers should be more comprehensive and would help the reader to gain a better understanding of the selection procedures for modified aptamers.  The authors discuss antisense oligonucleotide-based therapies which is out of scope for this paper so I would recommend that the authors change the title and introduction to reflect both aptamers and antisense oligonucleotides.  I also think that the authors should discuss the selection procedures for modified aptamers in more detail.  Not just focus on SONOMers and spiegelmers.      

Before I can recommend publication, the following points should be addressed.   

Line 37 “of” change to ”for”

“Section 2 Types of Oligonucleotide,  Antisense Oligonucleotides”:  This section seems out of place for this review as you are focusing on aptamers in your introduction.  I would recommend you change the title and introduction to include antisense oligonucleotides which are not aptamers.  This is important because the definition of an aptamer is that it binds to the antigen based on the 3D structure whereas antisense oligonucleotides bind based on their complementary base pairing. 

Line121:  Even PCR efficiency using specialist polymerases suffers from poor PCR amplification. 

Line 125:  Can the authors discuss the difference in selection procedures between the normal SELEX protocol and the selection procedure for SOMAmers and the selection of Spiegelmers.  This is since the SOMAmers and Spiegelmers feature very heavily in this review.  Also, the authors should discuss the selection of modified XNA aptamers. 

The authors mentioned that the transcription efficiency of modified aptamers is low which is a problem in their selection.  The authors should discuss how to sequence modified aptamers?

Figure 1:  The authors show Cell SELEX in the figure but make no reference to CELL-SELEX in the text.  The authors should discuss the differences in selection procedure between these two types of selection.   The authors should discuss the problems with Cell-SELEX, i.e the poor specificity and difficulty in characterizing where exactly the aptamer binds on the cell.

Line 214:  You mean aptamers can be conjugated to nanoparticles?

Line 572:  Not just proteins,  small molecules also.  

Author Response

Reviewer Comments: Reviewer #1 The Review discusses the current state of the art in therapeutic aptamers and antisense oligonucleotides in the diagnosis and treatment of hematological diseases.  The manuscript is well written and would be a useful source in the area in terms of describing the aptamers that have been developed in this area, the format and focus of the review need to be revised.  For instance, the discussion of the methods for the development of modified aptamers should be more comprehensive and would help the reader to gain a better understanding of the selection procedures for modified aptamers. The authors discuss antisense oligonucleotide-based therapies which is out of scope for this paper so I would recommend that the authors change the title and introduction to reflect both aptamers and antisense oligonucleotides.  I also think that the authors should discuss the selection procedures for modified aptamers in more detail.  Not just focus on SONOMers and spiegelmers.        Before I can recommend publication, the following points should be addressed.  

Comment 1: Line 37 “of” change to ”for”

Response to Comment 1: On line 37, “of” has been changed to “for”.  

Comment 2: “Section 2 Types of Oligonucleotide, Antisense Oligonucleotides”:  This section seems out of place for this review as you are focusing on aptamers in your introduction.  I would recommend you change the title and introduction to include antisense oligonucleotides which are not aptamers.  This is important because the definition of an aptamer is that it binds to the antigen based on the 3D structure whereas antisense oligonucleotides bind based on their complementary base pairing.

Response to Comment 2: We thank the Reviewer for these suggestions. As we have extensively discussed about antisense oligonucleotides and we have dedicated a table for their use in clinical practice, we have changed the title as follows “Aptamers and antisense oligonucleotides in diagnosis and treatment of hematological diseases”. In addition, we have introduced this term in the introduction (line 48).  

Comment 3: Line121:  Even PCR efficiency using specialist polymerases suffers from poor PCR amplification.

Response to Comment 3: On line 122, the sentence was rephrased as follows “however, this latter allows selection of high affinity aptamers, but transcriptional yield is very low also because of poor PCR amplification.”  

Comment 4: Line 125:  Can the authors discuss the difference in selection procedures between the normal SELEX protocol and the selection procedure for SOMAmers and the selection of Spiegelmers.  This is since the SOMAmers and Spiegelmers feature very heavily in this review.  Also, the authors should discuss the selection of modified XNA aptamers.

Response to Comment 4: SOMAmers are selected through SELEX process and SOMAmers differ from standard aptamers only for modified oligonucleotides used. On line 125, the sentence was rephrased as following “these modified single-stranded aptamers or Slow Off-rate Modified Aptamers (SOMAmers) bind to target molecules during SELEX process with less polar and more hydrophobic interactions”.

On line 178, the following text regarding Spiegelmer selection was added: “Spiegelmers are selected through SELEX using the synthetic mirror image of the natural target for positive selection. Bound aptamers are in the natural D-configuration and are later amplified, cloned, and sequenced by stereoselective enzymes. Finally, the produced aptamer sequences are employed for synthetization of L-aptamers using enantiomeric (L-)ribonucleotides [25].”

On line 183, the following discussion on XNA aptamers has been added “Finally, xenobiotic nucleic acid (XNA) libraries is the most recent approach to markedly increase aptamer stability [28]. XNAs are artificial and are different from natural DNA and RNA because of the presence of alternative backbone or sugar congeners making XNAs highly resistant to nuclease degradation [28-30]. XNA aptamers are then selected through cross-chemistry X-SELEX approach using immobilized targets on solid phase or gel, and subsequently selected XNA aptamers are reverse transcript, amplified, or sequenced [30]. To date, XNA aptamers are available for targeting hen-egg lysozyme, HIV trans-activating response RNA element and HIV-reverse transcriptase, human thrombin, human vascular endothelial growth factor 165, and human neutrophil elastase [28-30]. However, in vitro XNA selection is more challenging compared to other aptamers [29].”  

Comment 5: The authors mentioned that the transcription efficiency of modified aptamers is low which is a problem in their selection. The authors should discuss how to sequence modified aptamers?

Response to Comment 5: On line 115, the following text was added “amplified, cloned, and sequenced”.  

Comment 6: Figure 1:  The authors show Cell SELEX in the figure but make no reference to CELL-SELEX in the text.  The authors should discuss the differences in selection procedure between these two types of selection.   The authors should discuss the problems with Cell-SELEX, i.e the poor specificity and difficulty in characterizing where exactly the aptamer binds on the cell.

Response to Comment 6: The following text has been added on line 128: “The Cell-SELEX process is another way for selection of aptamers using cell lines (Figure 1B). In particular, ssDNA libraries are incubated with cells known to express targeted protein (e.g. CD30 on Hodgkin Lymphoma cell lines), and sequences bind with various affinity targeted protein. Unbound sequences are washed away, while bound sequences are first eluted by heating and subsequently incubated with cells not expressing targeted proteins for counterselection. Afterwards, unbound sequences which do not recognize non-specific targets on negative cells are later reverse transcript and amplified by PCR [3]. However, Cell-SELEX requires suitable cell lines expressing or not the targeted proteins for positive and negative selection, respectively, and aptamer specificity remains lower than that obtained by SELEX process [4].”  

Comment 7: Line 214:  You mean aptamers can be conjugated to nanoparticles?

Response to Comment 7: On line 214, the text has been changed as follows “aptamers can be conjugated to nanoparticles”.  

Comment 8: Line 572:  Not just proteins,  small molecules also.

Response to Comment 8: On lines 572-573, the text has been adjusted as following “specifically recognize proteins and small molecules”.  

Reviewer 2 Report

I finished evaluating the article “Aptamers in diagnosis and treatment of

hematological diseases” by Valentina Giudice et al. The article reviews recent advances in preclinical and clinical application of aptamers in malignant and non-malignant hematological diseases incuding: Sickle Cell Disease, Aplastic Anemia, Graft versus Host Disease, Multiple Myeloma, Lymphomas, and Leukemias. The work overviews basic information and importance of anti-sense oligonucleotide and aptamers technology.  The topic is very interesting, modern and fits to the scope of Int. J. Mol. Sci.. The extent and structure of review is adequate and logic respectively. The review was clear, concise and in general, well written in general, the only remark I have is the following:

  1. The credit to the two original aptamer papers should be provided:

Tuerk C, Gold L. "Systematic evolution of ligands by exponential enrichment: RNA ligands to bacteriophage T4 DNA polymerase". Science and  Ellington AD, Szostak JW. "In vitro selection of RNA molecules that bind specific ligands". Nature.

Author Response

Reviewer #2 I finished evaluating the article “Aptamers in diagnosis and treatment of hematological diseases” by Valentina Giudice et al. The article reviews recent advances in preclinical and clinical application of aptamers in malignant and non-malignant hematological diseases incuding: Sickle Cell Disease, Aplastic Anemia, Graft versus Host Disease, Multiple Myeloma, Lymphomas, and Leukemias. The work overviews basic information and importance of anti-sense oligonucleotide and aptamers technology.  The topic is very interesting, modern and fits to the scope of Int. J. Mol. Sci.. The extent and structure of review is adequate and logic respectively. The review was clear, concise and in general, well written in general, the only remark I have is the following:  

Comment 1: The credit to the two original aptamer papers should be provided: Tuerk C, Gold L. "Systematic evolution of ligands by exponential enrichment: RNA ligands to bacteriophage T4 DNA polymerase". Science and  Ellington AD, Szostak JW. "In vitro selection of RNA molecules that bind specific ligands". Nature.

Response to Comment 1: We apologize for have missing these two references which have been added to the new version of the manuscript.

Round 2

Reviewer 1 Report

The authors have addressed all the points.  

Author Response

Comment: The authors have addressed all the points.

Response to Comment: We thank the Reviewer.